# Delta-like 1 and Delta-like 4 differently require their extracellular domains for triggering Notch signaling in mice

Ken-ichi Hirano[1], Akiko Suganami[2], Yutaka Tamura[2], Hideo Yagita[3], Sonoko Habu[3], Motoo Kitagawa[4], Takehito Sato[1], Katsuto Hozumi[1]*

[1]Department of Immunology, Tokai University School of Medicine, Isehara, Japan; [2]Department of Bioinformatics, Graduate School of Medicine, Chiba University, Chiba, Japan; [3]Department of Immunology, Juntendo University School of Medicine, Tokyo, Japan; [4]Department of Biochemistry, International University of Health and Welfare School of Medicine, Narita, Japan

**Abstract** Delta-like (Dll) 1 and Dll4 differently function as Notch ligands in a context-dependent manner. As these ligands share structural properties, the molecular basis for their functional difference is poorly understood. Here, we investigated the superiority of Dll4 over Dll1 with respect to induction of T cell development using a domain-swapping approach in mice. The DOS motif, shared by Notch ligands—except Dll4—contributes to enhancing the activity of Dll for signal transduction. The module at the N-terminus of Notch ligand (MNNL) of Dll4 is inherently advantageous over Dll1. Molecular dynamic simulation revealed that the loop structure in MNNL domain of Dll1 contains unique proline residues with limited range of motion. The Dll4 mutant with Dll1-derived proline residues showed reduced activity. These results suggest that the loop structure—present within the MNNL domain—with a wide range of motion ensures the superiority of Dll4 and uniquely contributes to the triggering of Notch signaling.

**\*For correspondence:**
hozumi@is.icc.u-tokai.ac.jp

**Competing interests:** The authors declare that no competing interests exist.

## Introduction

Notch system is highly conserved from invertebrates to mammals and regulates cell fate decisions during the development of various organs (*Bray, 2006*; *Kopan and Ilagan, 2009*). This system is composed of four Notch receptors (Notch1–4) and four Notch ligands, Delta-like (Dll) 1, Dll4, Jagged (Jag) 1, and Jag2 in mammals; the interaction between these receptors and ligands induces the proteolysis of the Notch receptor, resulting in the translocation of Notch intracellular domain (NICD) into the nucleus, where the binding of NICD with transcription regulators such as Rbpj and MamL1 activates several target gene transcription. It is well known that Notch system plays pivotal roles in the development of various mammalian tissues, including neuron/glia, intestine, lung, pancreas, and so on. Interestingly, the interaction between Notch receptors and ligands is different in each tissue.

The Notch system plays essential roles in T lymphopoiesis in the thymus. Loss-of-function experiments using Notch1-deficient hematopoietic progenitor cells (HPCs) and Dll4-deficient thymic epithelium in vivo clearly demonstrated that the Notch1-Dll4 interaction is pivotal for T lymphopoiesis in the thymus (*Radtke et al., 1999*; *Hozumi et al., 2008b*; *Koch et al., 2008*). It has been long believed that three-dimensional distinctive structure of the thymus was necessary for T lymphopoiesis. However, it has been found that T lymphopoiesis can be recapitulated in an in vitro monolayer-culture system, in which HPCs bear the exogenous active fragment of Notch1 or are cultured on Notch ligand-expressing stromal cells (*Hozumi et al., 2003*; *Schmitt and Zúñiga-Pflücker, 2002*). In this culture system, Dll1, as well as Dll4, can promote HSCs to differentiate into T lineage cells;

however, the efficiency of Dll1 to promote HSC differentiation into T cells is lower than that of Dll4 (*Besseyrias et al., 2007*; *Mohtashami et al., 2010*).

As Dll1 expression is not clearly detected in the mammalian thymus (*Heinzel et al., 2007*), and the Dll1-deficient thymic epithelium remained intact (*Hozumi et al., 2004*), this in vitro result seems not be worth paying attention to. However, from the evolutionary point of view, cartilaginous fish possesses only Dll1 but not Dll4, suggesting that Dll1 is the only Notch ligand which contributes to T lymphopoiesis in the thymus of cartilaginous fish (*Bajoghli et al., 2009*). As cartilaginous fish lack the development of some major T cell sub-populations in the thymus (*Venkatesh et al., 2014*), the acquisition of Dll4 seems to be associated with the explosive evolution of T cell-mediated immune system. Furthermore, it has been reported that in some processes, such as the development of arterial vascular endothelium and presomitic mesoderm, Dll1 is not fully substituted by Dll4 (*Preuße et al., 2015*; *Tveriakhina et al., 2018*). Taken together, the physiological function of Dll1 and Dll4 is different. However, the molecular basis underlying this difference, and the preferential contribution of Dll4 to T lymphopoiesis is not fully understood.

Dll1 and Dll4 are type I cell-surface proteins that are composed of several distinct domains; MNNL (module at the N-terminus of Notch ligands), DSL (Delta/Serrate/Lag-2), and 8 EGF-like repeats (*Kopan and Ilagan, 2009*). A study that used recombinant soluble Notch1 and Dll proteins revealed that the N-terminus of Dll1/4 spanning from MNNL to the 3rd EGF-like repeat is enough to compare the simple affinity to Notch1, which was 10-fold higher for Dll4 than for Dll1 (*Andrawes et al., 2013*). Especially, the 1st and 2nd EGF-like repeat of Dll1 are known to form a distinct secondary structure. The DOS motif is common among Dll1, Jagged1, Jagged2—but not in Dll4—which were reported to contribute to binding with Notch receptors (*Kopan and Ilagan, 2009*; *Komatsu et al., 2008*). Therefore, it is assumed that Dll4 binds to Notch1 with stronger affinity than Dll1 and in a different manner from other Notch ligands, as far as the truncated and soluble form of Notch receptors and ligands are concerned.

The high-resolution structure of Notch1-Dll4 was revealed using a high-affinity mutant of Dll4, showing that Dll4 binds to Notch1 via two domains, MNNL and DSL (*Luca et al., 2015*). However, the contribution of the MNNL domain to the difference between Dll1 and Dll4 with the swapping strategy has not been reported. Recently, it has been shown that functional differences between Dll1 and Dll4, observed in vivo, resides in the extracellular domains (*Tveriakhina et al., 2018*). Moreover, these ligands differ in their ability to activate different Notch receptors, such as Notch1 and Notch2, in vitro, which is due to their N-terminal region that encompasses MNNL to the 3rd EGF-like repeat. However, their difference cannot simply be accounted for by interfacial residues in the MNNL-DSL region (*Tveriakhina et al., 2018*), which has been observed using high-resolution structural information (*Luca et al., 2015*); moreover, the part(s) of the domain, which have significant functions in the interaction with Notch1, remain to be known.

In this study, we confirmed the superior function of Dll4 compared to Dll1 as a transmembrane Notch ligand; Dll1 could not promote T cell development in the BM and Dll4 could deliver Notch1-mediated signaling more prominently than Dll1 in vitro. We constructed several domain-swapping mutants of Dll1/Dll4 and estimated the ability of each domain to induce T lymphopoiesis and Notch1-mediated signaling in vitro. Among the domains previously reported, we revealed that the MNNL domain of Dll4 contributes to the stable binding to Notch1, presumably via flexible loop structure with disulfide bond including several interfacial residues within the domain. In contrast, that of Dll1 contains a similar loop structure with limited range of motion. These results suggested that Dll1 and Dll4 differently bind with Notch1 due to the different dependency on their domain structures.

## Results

### Ectopic expression of Dll4 but not Dll1 induces T cell development in the bone marrow

We established conditional transgenic mice in which one copy of *Dll1* or *Dll4* gene was transcribed by CAG promoter after a Cre-dependent gene depletion of floxed *GFP* cDNA with the translational termination codon at *Rosa26* locus (hereafter referred to as iD1 for Dll1 and iD4 for Dll4 Tg mice, respectively; *Figure 1—figure supplement 1*). The expression of GFP could be observed in CD45[+]

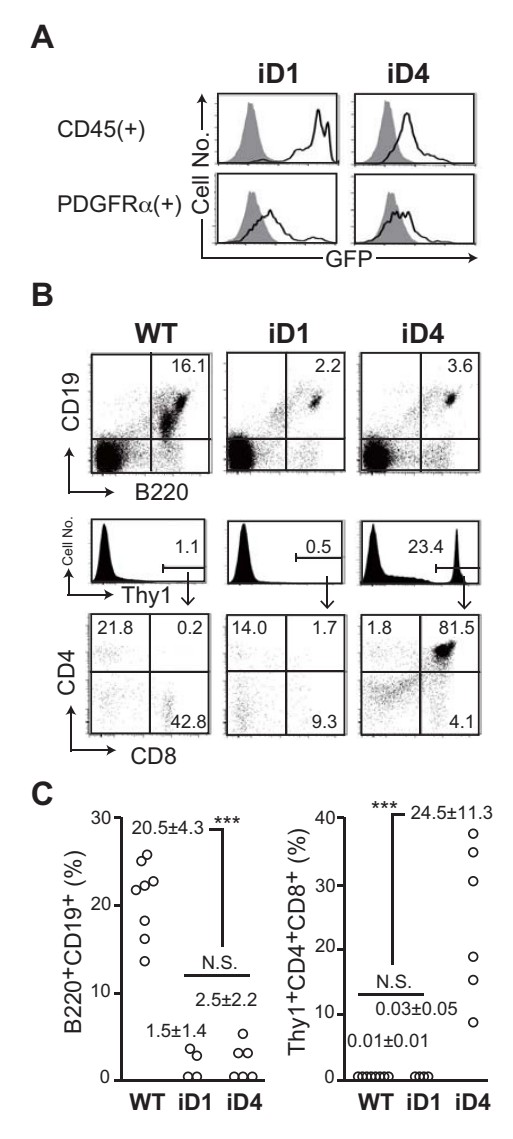

**Figure 1.** Effect of ectopic expression of Dll1 and Dll4 on the lymphopoiesis in the bone marrow. (A) GFP expression, which transcripts is driven by CAG promoter at the *Rosa26* locus of iD1 or iD4 mice, is detected in CD45+ hematopoietic or PDGFRα+ mesenchymal cell lineages in the bone marrow (BM) by flow cytometry. Open histograms indicate GFP expression of iD1 or iD4 mice, and filled histograms indicate the intrinsic fluorescence of the identical cell population of WT control mice. (B) Flow cytometry of the hematopoietic cells in BM obtained from tamoxifen-administrated WT, iD1, or iD4 mice with *Rosa^CreER* allele. One month after the last administration of tamoxifen, cells in BM were obtained and stained with mAbs against surface molecules as shown. Numbers in the profiles indicate the relative percentages, in Thy1+ cells (bottom, CD8 vs. CD4), for each quadrant or fraction. (C) The frequencies of B220+CD19+ B-lineage cells and Thy1+CD4+CD8+

*Figure 1 continued on next page*

hematopoietic and PDGFRα+ mesenchymal cells in the bone marrow (BM) (*Figure 1A*), although the expression of GFP in both cell lineages of iD1 mice were obviously higher than those of iD4 mice, which seemed to be due to the difference of sequences between the inserted *Dll1* and *Dll4* cDNA (*Figure 1—figure supplement 1*). These mice were bred with *Rosa^CreER* mice, and administered tamoxifen to systemically induce ectopic expression of Dll1 or Dll4. Up on the exogenous expression of Dll1 or Dll4 in the BM, substantial reduction in abundance of B220+CD19+ B-lineage cells was observed (*Figure 1B and C*), suggesting that Dll1 or Dll4-mediated Notch signaling occurred in hematopoietic progenitor cells (HPCs), and abrogated B cell development in the BM. However, ectopic appearance of Thy1+CD4+CD8+ immature T cells was detected only in the BM with exogenous Dll4 (*Figure 1B and C*), even though the expression level appeared to be lower. These results indicated that Dll4 can stimulate Notch signaling into HPCs more efficiently than Dll1, and that the amount of Notch signaling necessary for the inhibition of B cell development is lower than that required for the induction of T cell development in the BM.

## Dll4 promotes the growth and development of Thy1+CD25+ T-lineage cells on OP9 stromal cells from hematopoietic progenitor cells more efficiently than Dll1

To evaluate the potency of Dll1 and Dll4 at inducing T-lineage cells from HPCs on OP9 stromal cells (*Radtke et al., 1999*; *Hozumi et al., 2008a*) in vitro, we established the OP9 transfectants that expressed either Dll1 or Dll4 on the surface in response to doxycycline removal (Dox, Tet-off system). The expression of these proteins on the cell surface was quantitatively detected by intracellular staining with an isotype control or by an anti-HA mAb that recognizes the C-terminus HA epitope of both Dll1 and Dll4 in a dose-dependent manner (*Figure 2A*, left panels). This system enabled the direct comparison of the potencies of Dll1 and Dll4, and confirmed that Dll4 induced T-lineage cells on OP9 cells more efficiently than Dll1 did from HPCs as shown previously (*Besseyrias et al., 2007*; *Mohtashami et al., 2010*). In every comparable situation, for similar induction of Thy1+CD25+ T-lineage cells, lower surface expression of Dll4 was required (*Figure 2A*). Moreover, by using serial dilution of Dox, we determined the EC50 values of Dll1 or Dll4 expression in OP9 cells for the induction of CD25 during the initiation of T cell differentiation

*Figure 1 continued*

immature T cells among the total cells of BM were examined as shown in B (mean ± SD; WT, n = 8; iD1, n = 4; iD4, n = 6; ***, p<0.001; N.S.: not significant; unpaired Student's *t*-test).

The online version of this article includes the following source data and figure supplement(s) for figure 1:

**Source data 1.** Raw data used to generate the graph in *Figure 1C*.

**Figure supplement 1.** Establishment of conditional Dll transgenic mice by recombinase-mediated cassette exchange (RMCE).

(*Figure 2B*). Based on the results from three independent experiments, we estimated the $EC_{50}$ values at 16710 ± 1820 and 4217 ± 1594 dMFI for Dll1 and Dll4 (*Figure 2—source data 2*), respectively, and demonstrated that Dll4 in OP9 cells exhibited higher ability to induce T-lineage cells than Dll1.

## DOS motif present in the 1st/2nd EGF-like repeat (EGF1-2) of extracellular regions of Dll1, but not in those of Dll4, represents a unique structure and modifies the function of both Dll1 and Dll4

To investigate the molecular basis for the functional difference between Dll1 and Dll4, we generated several swapping chimeras of the DSL and the 1st and 2nd EGF-like repeats (EGF1-2, depicted E1-2 in *Figure 3*) containing DOS motif in Dll1 (*Figure 3A*) in order to determine whether the EGF1-2 of Dll4 that lacks the DOS motif will allow it to function better with Notch1. To evaluate that, we prepared the cell lines producing soluble or transmembrane Notch1/2 proteins with or without glycosyl modification by Lfng or Mfng, glycosyltransferase, because the fringe-modified Notch increases the affinity towards the Dll family members (*Figure 3—figure supplement 1*). The glycosyl modification of Notch receptors by Lfng or Mfng is well known to be effective in enhancing their affinities towards Dll but reduced their affinities to the Jag family members (*Fortini, 2000*; *Hicks et al., 2000*). Dll1-based chimeras with Dll4-derived EGF1-2, named D1-D4E1-2 and D1-D4DSL/E1-2 (*Figure 3A*), lost their ability to bind to Notch receptors (*Figure 3—figure supplement 1*). Next, the activity of signal transduction via Notch1 was examined by luciferase reporter assay (TP-1 reporter, driven by 6xRbpj binding sites) with or without Lfng. Expression levels of NotchL were monitored by the expression of GFP in this experiment and almost all transfectants were comparable. (*Figure 3—figure supplement 2*). While the high levels of NotchL expression used in this experiment made it less sensitive to the difference between Dll1 and Dll4, the effects of the domain-swapping chimeras were strongly evident. These chimeras exhibited reduced activity to transduce Notch1-mediated signaling in the reporter assay (*Figure 3B*). Importantly, reduced or eliminated activity for signal transduction by D1-D4DSL and D1-D4DSL/E1-2 was seen even though these constructs were expressed even more highly than Dll1. As only D1-D4E1-2 transfectant showed the lower expression of GFP (*Figure 3—figure supplement 2*), its weak activity in the reporter assay might be due to the insufficient expression. However, its binding efficiency with Notch1/2 (*Figure 3—figure supplement 1*) and signal transduction by D1-D4DSL/E1-2 in the reporter assay were also low. Thus, it was likely that the DOS motif present in EGF1-2 of Dll1 is necessary for the function of Dll1, which was consistent with the previous report (*Liu et al., 2017*). Moreover, these swapping mutants did not promote T cell differentiation in vitro (*Figure 3—figure supplement 1*). Conversely, Dll4-based chimeras with Dll1-derived EGF1-2, D4-D1E1-2, and D4-D1DSL/E1-2 enhanced their activities, binding to Notch1 (*Figure 3—figure supplement 1*), signal transduction via Notch1 (*Figure 3B*) and induction of T cell differentiation (*Figure 3—figure supplement 1*). These results indicated that the DOS motif also contributes to enhancing the activity of Dll4. Similar results were obtained with other swapping mutants containing one amino acid substitution—$P^{260}$ of Dll1 to N, which was conserved in the DOS motif, or vice versa—D1-P260N and D4-N258P (data not shown), as shown in our previous study (*Liu et al., 2017*). Moreover, the swapping chimeras of DSL domains neither efficiently bind to Notch receptors nor induce Notch signaling (*Figure 3B* and *Figure 3—figure supplement 1*), indicating their specificity to exert full activity. These results suggested that DOS motif, shared by NotchLs except Dll4, is crucial for the functionality of Dll1 and even capable of enhancing that of Dll4 in chimera. Thus, the superiority of Dll4 cannot be explained by the difference in DSL and EGF1-2 domains, which are known to be critical for binding to Notch receptors.

Further, we have established a hamster-derived mAb, HRJ1-5, by immunization with purified rat Jag1 protein, which was broadly cross-reactive with murine and human Dll1, Jag1, and Jag2, but not to Dll4 (*Figure 3C* and data not shown). To identify the epitope recognized by HRJ1-5, we validated

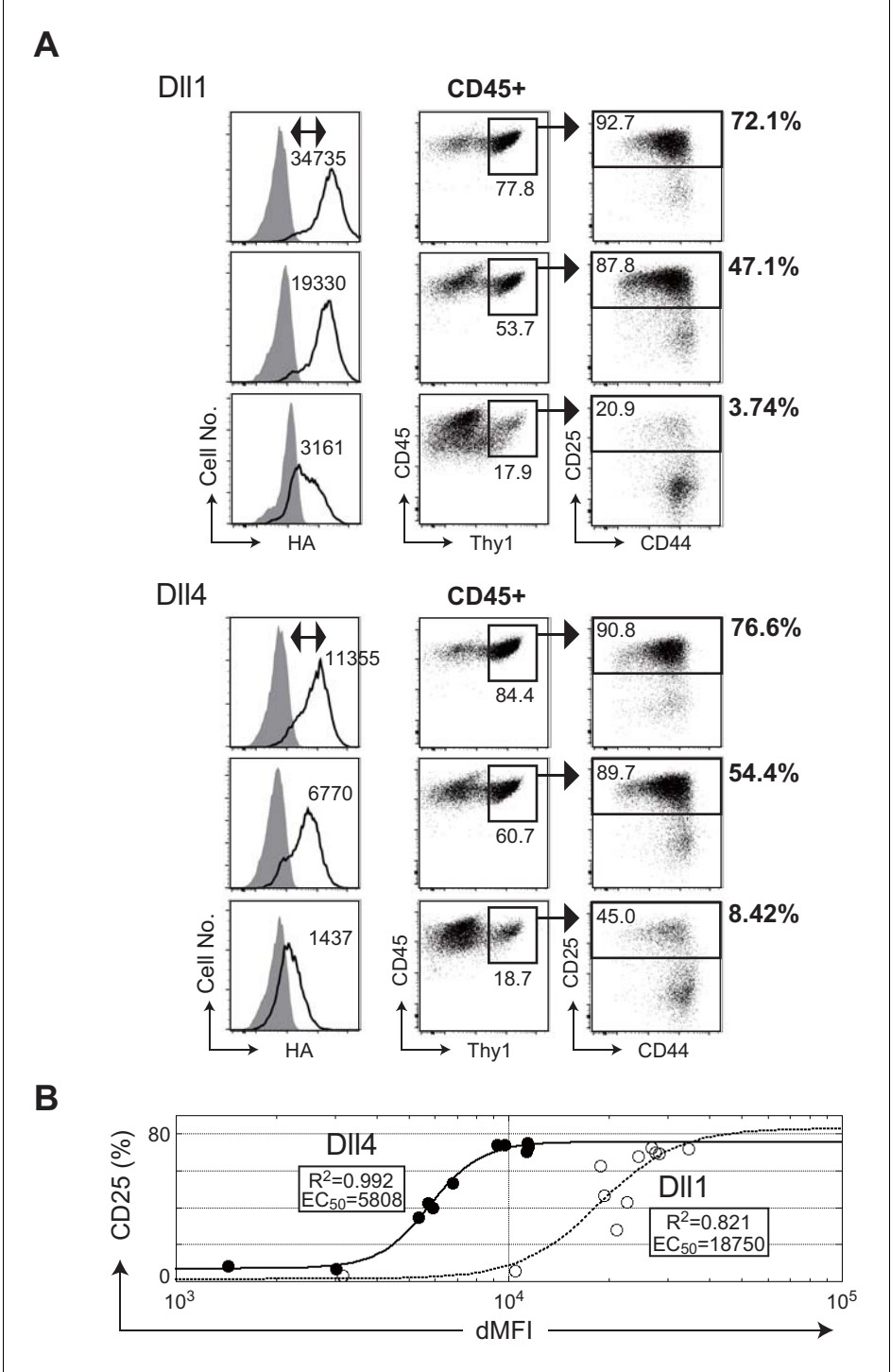

**Figure 2.** The efficiencies of T cell induction from hematopoietic progenitors by Dll1 or Dll4 on the monolayer cultures with OP9 stromal cells. (**A**) Serial induction of exogenous Dll1 or Dll4 labeled at the C-terminus with HA-epitope driven by Tet-off system in OP9 cells was detected by flow cytometry using anti-HA mAb for intracellular staining. OP9 transfectants were treated with doxycycline (0.01, 0.8, and 0.3 ng/mL) for Dll1 or Dll4 to suppress the full activation of their transcription. Open histograms indicate anti-HA mAb staining (Dll1 or Dll4), and filled histograms indicate staining with control rabbit IgG. The difference of each MFI (dMFI) is shown in the left panels. E14.5 fetal liver-derived lineage markers (Gr-1, CD11b, TER119, and CD19)-negative c-kit-positive hematopoietic progenitors were cultured on the Dll1- or Dll4-bearing OP9 cells for 7 days and stained with lineage markers for analysis (Gr-1, CD11b, ST2, and DX5) and (CD45, CD19, Thy1, CD44, and CD25). CD45+ and CD45+Thy1.2+ cells

*Figure 2 continued on next page*

*Figure 2 continued*

were analyzed in the center and right panels, respectively. The frequencies (%) of lineage markers-negative CD19⁻Thy1⁺CD25⁺ T-lineage cells in CD45⁺ live cells are shown in the right side of the panels. Numbers in the dot-plot represent the relative percentages for each corresponding fraction. (B) The effectiveness of serial expression (dMFI) of Dll1 (open circle, dotted line) or Dll4 (closed circle, solid line) on OP9 cells for the induction of CD25⁺T-lineage cells (CD25) shown in A was evaluated by logistic regression analysis. In a 4-parameter logistic equation, $EC_{50}$ and $R^2$ values were calculated and shown in the graph.

The online version of this article includes the following source data for figure 2:

**Source data 1.** Raw data used to generate the graph in *Figure 2B*.
**Source data 2.** Measurement of the induction efficiencies for the CD25⁺ T-lineage cells of serial expression of Dll4 or Dll1 as $EC_{50}$ as shown in *Figure 2B*.
**Source data 3.** Raw data used to generate *Figure 2—source data 2*.

---

the reactivity of this mAb to the chimeras (*Figure 3D*). HRJ1-5 clearly bound to mutants with Dll1-derived EGF1-2 with DOS motif, indicating that the DOS motif contributes to the maintenance of the unique structure recognized by the antibody, which is critical for their function as NotchL.

## The MNNL domain of Dll4 is necessary for its function and is related to the superiority of Dll4 over Dll1

Previous reports have indicated that the truncated proteins of Dll family members from the N-terminus to the 3rd EGF-like repeat in the extracellular region are sufficient to trigger the Notch signaling (*Andrawes et al., 2013*; *Liu et al., 2017*), and that Dll4 directly binds to Notch1 via its MNNL and DSL domains (*Luca et al., 2015*). We next examined the significance of MNNL domain of Dll4 using other swapping chimera constructs (*Figure 4A*), which included HA tag for comparing protein expression levels. This shows that the chimeras are expressed to similar absolute levels per cell. Furthermore, swapping the MNNL domain in Dll1 does not affect expression of its DOS motif as detected by HRJ1-5, although swapping the MNNL domain of Dll4 does appear to reduce its reactivity with an antibody against the Dll4 DSL domain (*Figure 4—figure supplement 1*). Interestingly, Dll4-based chimera with Dll1-derived MNNL domain (D4-D1MNNL) reduced its binding to soluble Notch1 and Notch2, while Dll1-based chimera with Dll4-derived MNNL region (D1-D4MNNL) showed comparable activity to that of Dll1 (*Figure 4B*). Similarly, only D4-MNNL lost its activity to induce Notch1-mediated signaling with or without Lfng in the luciferase reporter assay (*Figure 4C*). Moreover, D4-D1MNNL failed to inhibit B cell development and support T cell differentiations in vitro (*Figure 4D*). These results demonstrated that the MNNL domain of Dll4 is necessary for its function as NotchL, and that there might be some difference(s) in MNNL domain of Dll1 and Dll4, which is responsible for the superiority of Dll4 over Dll1 for T cell induction.

## Molecular dynamic simulation enables different characterization of structural dynamics in MNNL domain

In order to determine the significance of the MNNL domain of Dll4 in forming a complex with Notch1, we constructed Dll/Notch1 complexes (Dll4/Notch1, depicted in *Figure 5A* and D4-D1MNNL/Notch1) and performed molecular dynamic simulations using these complexes. Consistent with our in vitro and in vivo experiments, the interaction energy of Dll4/Notch1 (*Figure 5B*, black line) was lower than that of D4-D1MNNL/Notch1 (*Figure 5B*, red line), suggesting that the former interaction was more stable. Then, we focused our attention on the differences in the amino acid sequence of the loop structure with disulfide bond (C-C loop, dotted circle, *Figure 5A and D*) present in the MNNL domain, between Dll1 and Dll4 (*Figure 5C*). This is because the C-C loop of Dll1 contains a characteristic proline-rich amino acid region (PEPP-region, underline, *Figure 5C*) that restricts the movement of the C-C loop in the MNNL domain (*Figure 5D*), probably due to the conformational rigidity created by the presence of several proline residues (magenta sticks, lower panels in *Figure 5D*). This loop structure also includes the residues (⁶⁴His and ⁶⁵Phe, bold green, *Figure 5C*) that comprise the binding surface of MNNL domain of Dll4 which interacts with Notch1 (*Luca et al., 2015*). As shown in *Figure 5E*, it was revealed that the movement of the C-C loop in the MNNL domain of Dll1 (red line) is completely restricted in comparison with that of Dll4 (black

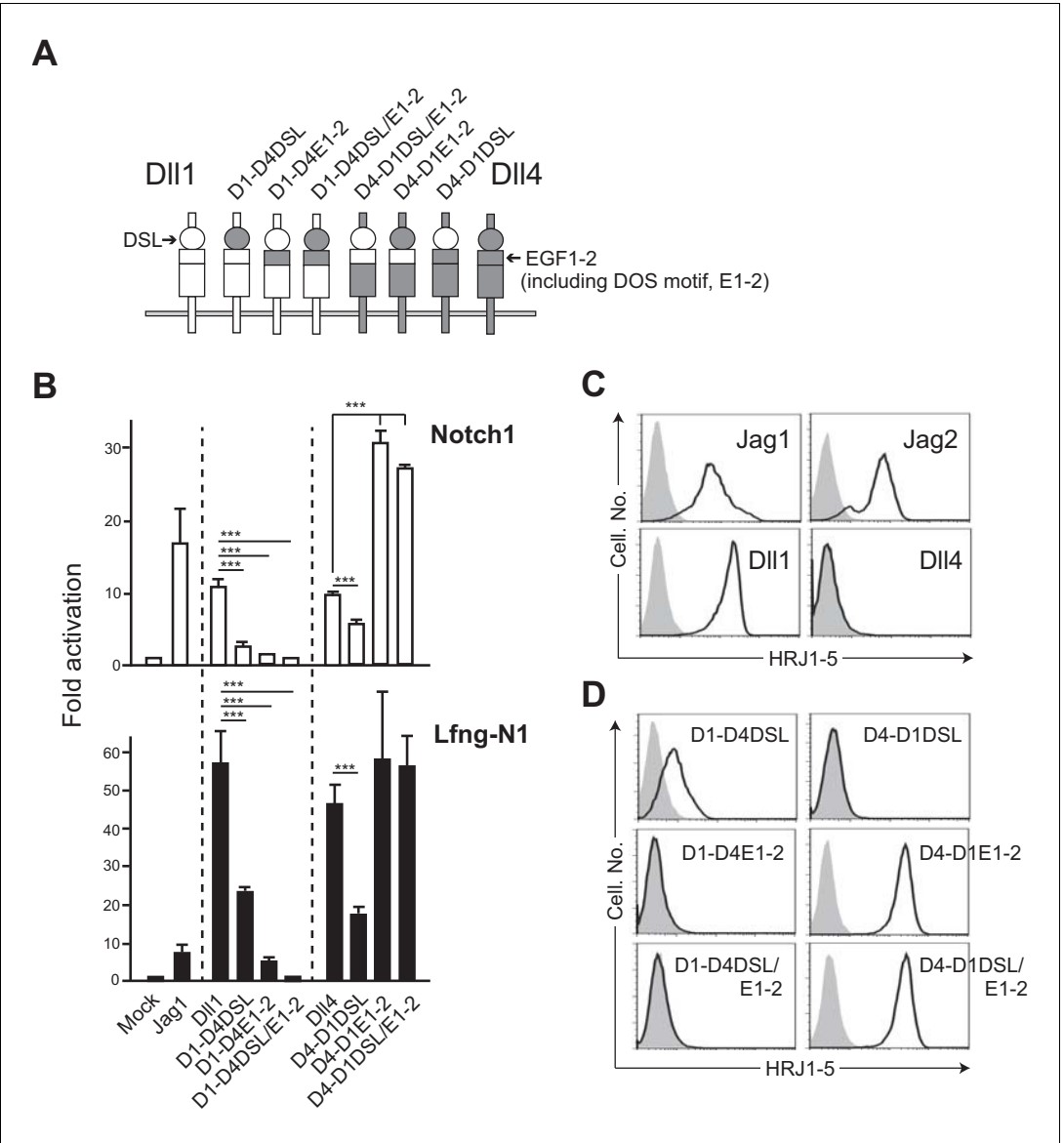

**Figure 3.** The swapping chimeras of DSL domain and/or the 1st/2nd EGF-like repeats between Dll1 and Dll4. (**A**) Schematic structure of the Dll variants. Dll1 and Dll4 were intact and depicted by open (Dll1) or filled (Dll4) columns. The DSL domain and the 1st/2nd EGF-like repeats (E1-2) were represented by circle and top region of square, respectively. D1-D4DSL, D1-D4E1-2, and D1-DSL/E1-2 were Dll1-based chimeras with Dll4-derived DSL domain and/or E1-2. Similarly, Dll4-based chimeras were generated with Dll1-derived domains (D4-D1DSL, D4-D1E1-2, and D4-D1DSL/E1-2). Expression of NotchLs were monitored by GFP expression (*Figure 3—figure supplement 2*). (**B**) The swapping chimeras of DSL domain and/or the 1st/2nd EGF-like repeats transduce Notch signaling. Stable transfectants expressing murine Notch1 and control vector (Notch1, open column) or Notch1 and Lfng (Lfng-N1, filled column) were transiently transfected with a TP1-luciferase reporter plasmid, pGa981-6, and a pRL-TK plasmid for internal control. Cells were harvested at 24 hr after transfection, and co-cultured for an additional 40 hr with the transfectants expressing the DSL/E1-2 swapping chimeras. The relative induction of luciferase activity in each sample (mean ± SD, n = 3; ***, p<0.001; unpaired Student's *t*-test) was calculated and described as fold activation against the control (value from the culture with mock transfectant not expressing Notch ligand). Data represents three independent experiments. (**C**) Monoclonal antibody originally established by us, HRJ1-5, broadly reacted with murine Jag1, Jag2, and Dll1, but not with Dll4. The transfectants of BM-derived mesenchymal cell line, originally not expressing any Notch ligands, established in our lab, expressing murine Jag1, Jag2, Dll1, or Dll4, were stained with HRJ1-5, and analyzed by flow cytometry. Open histograms indicate HRJ1-5 staining and filled histograms indicate staining with control hamster IgG. (**D**) Reactivity of HRJ1-5 with Dll chimeras. Each transfectant shown in the panel was stained with HRJ1-5 and analyzed as in C.

*Figure 3 continued on next page*

*Figure 3 continued*

The online version of this article includes the following source data and figure supplement(s) for figure 3:

**Source data 1.** Raw data (Fold activation) of luciferase activity used to generate the graph in *Figure 3B*.
**Figure supplement 1.** Characterization of the swapping chimeras of DSL and/or the 1st/2nd EGF-like repeats (EGF1-2) between Dll1 and Dll4.
**Figure supplement 2.** NotchLs transfectants used in *Figure 3* were established via infection with a retrovirus encoding the respective NotchL or their chimeric molecules of parental fibroblast cell lines (parent), and were monitored via GFP expression by flow cytometry.

line). This difference was shown in the movies (Dll4, *Figure 5—video 1*; Dll1, *Figure 5—video 2*). Therefore, we presumed that the restricted movement of the C-C loop in the MNNL domain of Dll1 prevents the inducible binding with Notch1. However, in the MNNL domain of Dll4, the flexible movement of the C-C loop is retained, which enables Dll4 to interact with Notch1 via the key residues (*Figure 5C and D*, *Figure 5—video 1*, *Figure 5—video 2* and *Figure 5—figure supplement 1*).

Furthermore, we constructed a Dll4-based mutant with a Dll1-derived PEPP-region (Dll4-PP, *Figure 5C*) to examine the contribution of the C-C loop in the MNNL domain to the complex formation with Notch1. The interaction energy of Dll4-PP/Notch1 and the movement of the C-C loop in the MNNL domain of Dll4-PP (*Figure 5B and E*, blue lines) were nearly identical to those of D4-D1MNNL or Dll1 (*Figure 5B and E*, red lines). This difference was also found in the movie (*Figure 5—video 3*). Thus, these results revealed that the flexible movement of the C-C loop in the MNNL domain of Dll4 plays an important role in complex formation with Notch1.

## The loop structure in MNNL domain of Dll4 with a wide range of motion contributes to efficient signal transduction

To confirm the significance of the dynamic motion of the C-C loop in MNNL domain, we compared the Notch binding and activating abilities of Dll4-PP—which retains two unique proline residues— with those of Dll4 (*Figure 5C*). Consistent with the simulation analysis, Dll4-PP showed weaker ability to bind to soluble Notch receptors (*Figure 6A*) and to transduce Notch signaling in the reporter experiment (*Figure 6B*), which were similar to those observed on using the Dll4-based chimera with Dll1-derived MNNL domain, D4-D1MNNL (*Figure 4B–D*). The strong reduction in functional activity of Dll4-PP was seen despite virtually unchanged levels of protein expression (*Figure 6—figure supplement 1*). These results suggested that the C-C loop in MNNL domain of Dll4 with a wide range of motion uniquely contributes to the triggering of Notch signaling.

## Discussion

In this study, we focused on the difference between Dll1 and Dll4, and revealed that Dll family members bind to Notch and trigger the signaling differently based on the structural features: DSL plus DOS motif for Dll1 and DSL plus MNNL for Dll4. The MNNL of Dll1 loses the ability to move widely with its rigidity and the EGF1-2 of Dll4 lacks the DOS motif. It has been well known that the DSL domain is highly conserved from invertebrates to mammals and is essential to maintain the structure and function of NotchLs (*Tax et al., 1994*; *Cordle et al., 2008*; *Chillakuri et al., 2012*; *Kershaw et al., 2015*). Moreover, the additional domain, MNNL in Dll4 and EGF1-2 in Dll1, is further required for the stable interaction with Notch. We showed here that Dll1 and Dll4 differently use the neighboring domains of DSL. However, these results were mainly obtained in vitro and might be seen under conditions in the middle of the dynamic range between Notch and NotchL, where the additional domains seem to be essential. Therefore, our conclusions need to be verified in physiological condition.

Molecular basis of the interaction between Notch1 and Dll4 was provided by the high-resolution structural analysis and showed that several interfacial residues within the MNNL and DSL domains of Dll4 directly interact with Notch1 (*Luca et al., 2015*). However, it was reported that the Dll4-based mutant containing an exchange of the MNNL from 65th Phe (Dll4) to Tyr (Dll1) (*Figure 5C*, bold green), and of DSL with several contact residues from Dll1, did not alter its selectivity toward Notch1 characteristic of Dll4. Therefore, it was suggested that residues outside of the MNNL-DSL contact

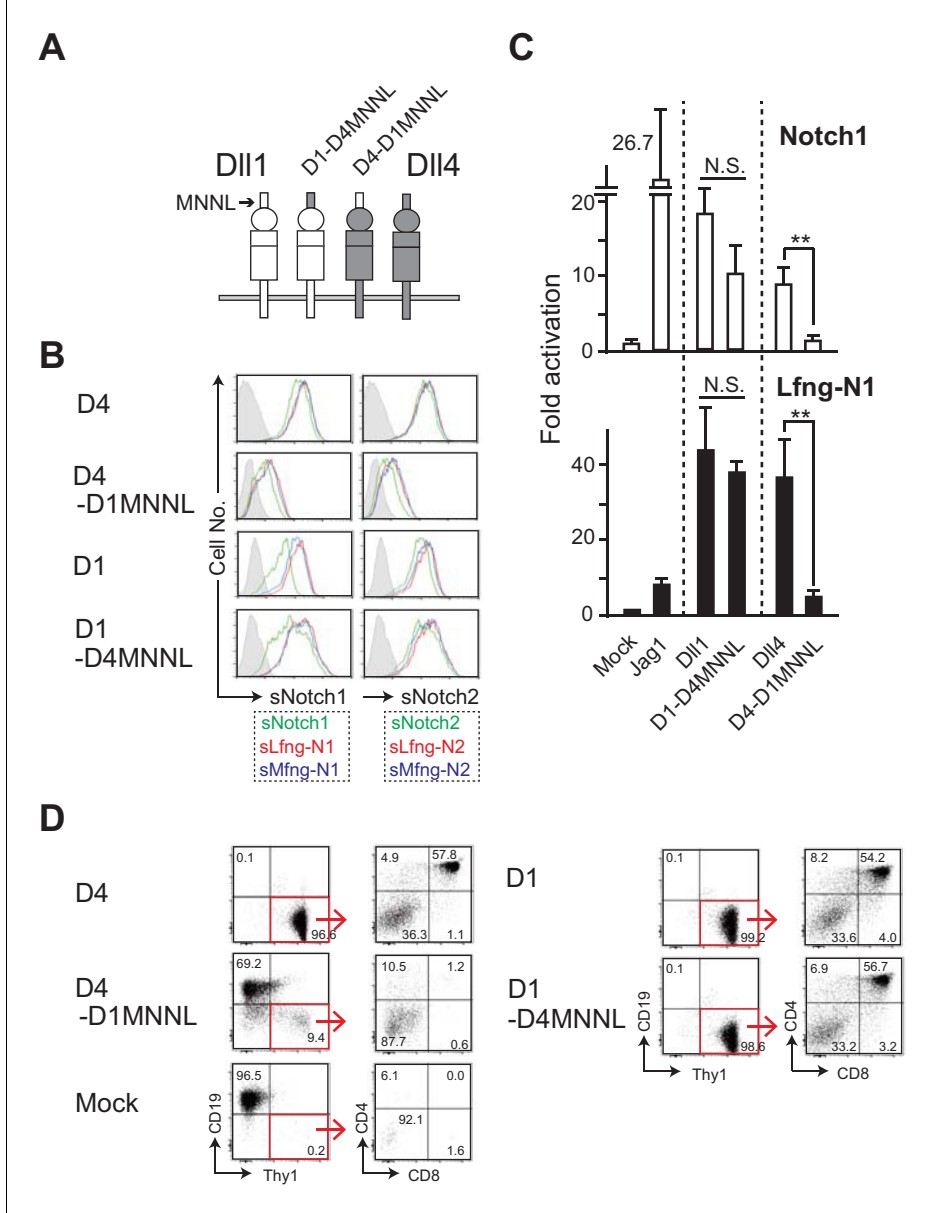

**Figure 4.** The swapping chimera of MNNL domain between Dll1 and Dll4. (**A**) Schematic structure of the Dll variant of MNNL (the N-terminal) domain. D1-D4MNNL and D4-D1MNNL were Dll1- and Dll4-based chimeras with Dll4- and Dll1-derived MNNL domains, respectively. Expression of NotchLs were monitored by the GFP expression, the intracellular staining with anti-HA mAb and the surface staining with HRJ1-5 or anti-Dll4 mAbs (**Figure 4—figure supplement 1**). (**B**) Binding activity of the MNNL swapping chimeras with soluble Notch1 and Notch2. Soluble Notch receptors comprised the N-terminus to the 15th EGF repeats and the Fc domain of human IgG1. The transfectants of BM-derived mesenchymal cell line expressing Dll1 (D1), Dll4 (D4), or their MNNL swapping chimeras (D4-D1MNNL and D1-D4MNNL) were incubated with soluble Notch1 (sN1) or Notch2 (sN2), and their binding was detected by anti-human IgG Ab as described in the Materials and methods. Soluble Notchs were produced by CHO cells expressing murine Lfng (red line), Mfng (blue line), or vector control (green line), and used in this experiment. (**C**) The MNNL swapping chimeras transduce Notch signaling. Reporter assay was carried out as shown in **Figure 3B**. The relative induction of luciferase activity in each sample (mean ± SD, n = 3; **, p<0.01; N.S.: not significant; unpaired Student's t-test) was calculated. Data represents three independent experiments. (**D**) Induction of T-lineage cells by MNNL swapping chimeras in in vitro. Fetal liver-derived lineage markers-negative c-kit-positive hematopoietic progenitors were cultured on the OP9 cells expressing Dll1, Dll4, or their MNNL chimeras for 13 days. After the cultures, live cells were stained with CD19, Thy1, CD4, and CD8, and

*Figure 4 continued on next page*

*Figure 4 continued*

analyzed by flow cytometry. Numbers in the dot-plot represent the relative percentages for each corresponding fraction or quadrant in Thy1$^+$CD19$^-$ fraction (red square).

The online version of this article includes the following source data and figure supplement(s) for figure 4:

**Source data 1.** Raw data (Fold activation) of luciferase activity used to generate the graph in *Figure 4C*.

**Figure supplement 1.** NotchLs transfectants used in *Figure 4* were monitored by the expression of GFP and the staining with anti-HA , anti-Dll4 or HRJ1-5 mAbs.

interface of Dll4 contribute to the functional characteristics of Dll4 (*Tveriakhina et al., 2018*). Alternatively, in this study, we demonstrate that, when approaching Notch1, the MNNL of Dll1 does not behave as that of Dll4 and that Dll1 efficiently triggers Notch1 depending on the DOS motif present in EGF1-2 of Dll1 but not Dll4. Thus, the unique approach of Dll family members is based on the domains used differently in the binding to Notch1, but not on the distinctive amino acid residues.

We observed the difference between ectopic Dll1 and Dll4 in the ability to induce and suppress T- and B-lineage cells, respectively, in the BM, where exogenous Dll proteins were expressed, at least, in both hematopoietic and mesenchymal cell lineages. Previous studies with retrovirus-mediated transduction of Dll1 or Dll4 into hematopoietic cells and transfer into lethally irradiated mice suggested the superiority of Dll4 over Dll1; however, these experiments did not ensure the sufficient expression of Dll1 to function (*de La Coste et al., 2005*). In our established mouse model, exogenous Dll1, as well as Dll4, clearly suppressed B cell development in the BM; however, it was not enough for the appearance of CD4, CD8-bearing immature T cells. These results suggested that Dll4 in BM transduces the Notch signaling into HPCs for T cell development more efficiently than Dll1 and confirmed that there is a different requirement regarding the threshold of Notch signaling for the suppression of B cell and the induction of T cell lineages in vivo, as previously shown in vitro (*Koga et al., 2018*). We have previously shown that on the monolayer culture of HPCs, Gata3 is necessary only for the induction of the T-lineage cells, but not for the suppression of B-lineage cells by Notch signaling, suggesting the presence of different mechanisms downstream of Notch signaling (*Hozumi et al., 2008b*). The quantitative difference in Notch signaling seems to be converted into the distinctive cellular events at the hematopoietic progenitor stage.

Although it has been suggested that Dll4 induces Notch1-mediated signaling more efficiently than Dll1, its quantitative evaluation was only conducted by their affinities using soluble forms of both Dll and Notch1 proteins as truncated parts of the extracellular domains and demonstrated that Dll4 binds with more than 10-fold higher affinity than Dll1 (*Andrawes et al., 2013*). As NotchL triggers Notch signaling in the neighboring cells only as a transmembrane protein and not in its truncated soluble form, alterations in its levels should be quantitatively evaluated only as a transmembrane protein. In this study, we aimed to compare full-length Dll proteins as transmembrane form by measuring their serially diluted expression on stromal cells and by their ability to induce T-lineage cells. Further, we evaluated their biological effect on HPCs. Dll4 present on stromal cells supported T cell differentiation 3–6-fold more efficiently than Dll1, which seemed to be less when compared to the difference in their affinities to Notch1. HPCs express both Notch1 and Notch2 on their surface, and the latter contributes to the induction of T-lineage cells only on interacting with Dll1 (*Besseyrias et al., 2007*; *Fiorini et al., 2009*), indicating that the total signaling induced by Dll1 was mediated by Notch1 as well as Notch2, while that induced by Dll4 was through Notch1 alone. We realized that the functional difference in the ability to induce T-lineage cells should be assessed under this situation.

It is obvious that the determination of T cell fate is regulated only by the *trans*-activation of Notch signaling between NotchL and Dll4 on thymic epithelium (environment), and Notch1 on thymic immigrants (progenitors). In contrast, Notch-NotchL interaction seems to facilitate not only *trans*-activation but also *cis*-inhibition during the development of various tissue, because Notch signaling occurs between two equivalent progenitors in which Notch and NotchL simultaneously function, according to the lateral inhibition theory (*del Álamo et al., 2011*; *Sprinzak et al., 2010*). In fact, functional difference between Dll1 and Dll4 is also observed during somitogenesis and Dll4 cannot rescue the defect of Dll1, which is explained by the difference between their action through the *cis*-inhibition (*Preuße et al., 2015*). This situation is not suitable due to its complexity and makes it difficult to

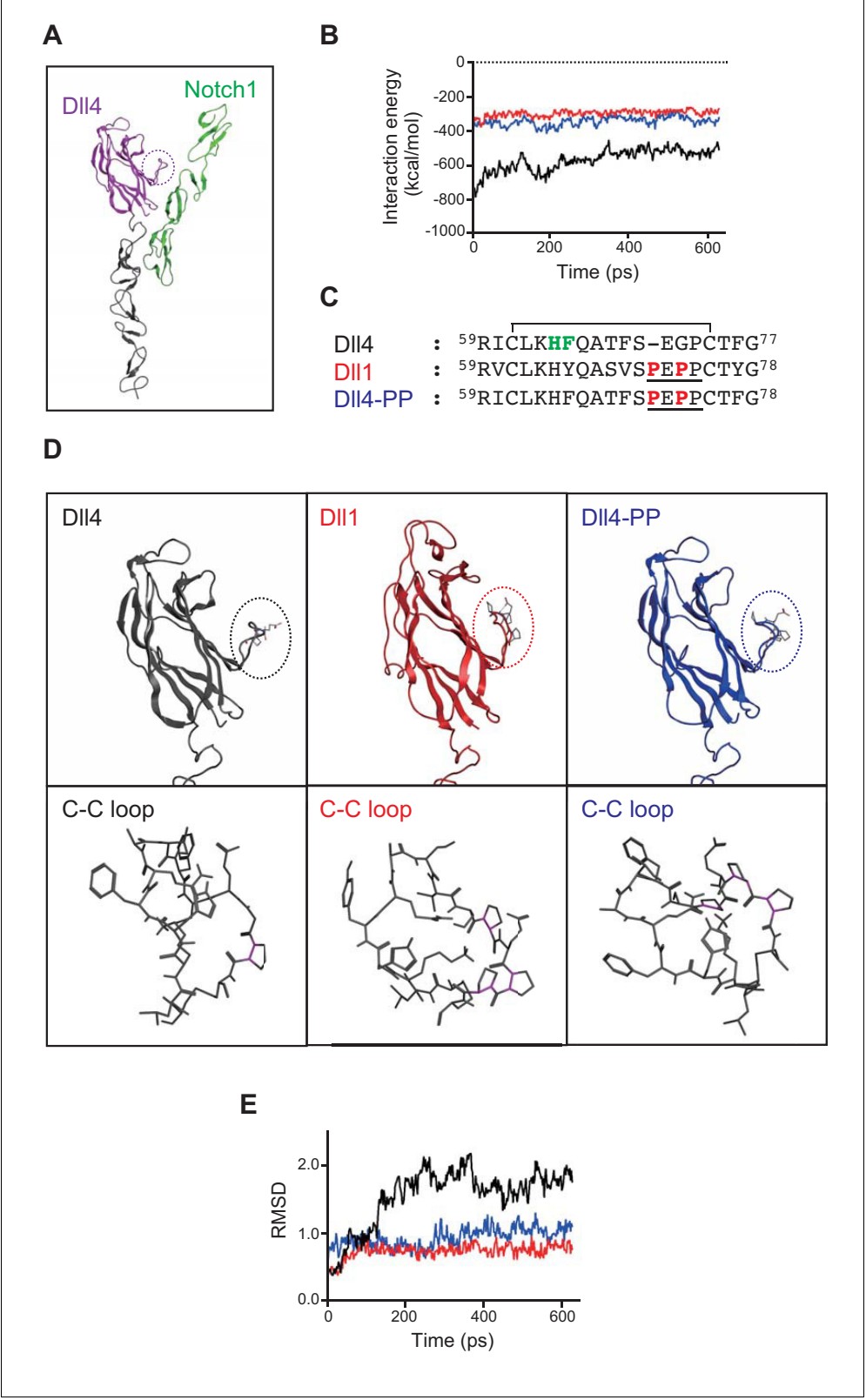

**Figure 5.** In silico analysis of Dll/Notch1 complexes. (**A**) Structure of Dll4 (MNNL, purple; DSL and EGF1-2, black) bound to Notch1 (EGF11-13, green) is shown in ribbon representation. The dotted circle indicates the loop structure with disulfide bond (C-C loop) in MNNL domain. (**B**) Molecular dynamic simulation of 600 ps for the interaction energy of Dll4/Notch1 (black), D4-D1MNNL/Nothc1 (red), and Dll4-based mutant with two proline residues (characteristic of Dll1) (Dll4-PP/Notch1, blue). (**C**) Amino acid (AA) sequence comparison of the C-C loop in MNNL domain between Dll4, Dll1, and Dll4-

*Figure 5 continued on next page*

Figure 5 continued

PP. Numbers on the AA sequences represent the position from the N-terminus. D4-PP is the Dll4-based mutant with the inserted (71st position) and substituted (73rd position) mutations of proline residue (bold red; characteristic of Dll1, underline). The AAs in the C-C loop contributing to the direct binding with Notch receptor are labeled (bold green) described as previously. Line over the sequence represents the disulfide bridge between cysteine residues (61st to 74th). (D) Structure of MNNL domain of Dll4 (black), Dll1 (red), and Dll4-PP (blue) in ribbon representation (upper panels) with enlarged wireframe model of the C-C loop (lower panels). Magenta, C-N bond in proline. (E) Molecular dynamic simulation of 600 ps for the RMSD of the C-C loop in the MNNL domain of Dll4 (black), Dll1 (red), and Dll4-PP (blue).

The online version of this article includes the following video and figure supplement(s) for figure 5:

**Figure supplement 1.** The loop structure with disulfide bond between 61Cys (C61) and 72Cys (C72) includes the key residues, 64His (H64) and 65Phe (F65), filled circle in *Figure 5C*, that comprise the binding surface of MNNL domain of Dll4 which interacts with Notch1.

**Figure 5—video 1.** Molecular dynamics simulation of Dll4 (*Figure 5—video 1*, black), Dll1 (*Figure 5—video 2*, red) and Dll4-PP (*Figure 5—video 3*, blue) with Notch1 (green) for 600ps were exhibited in the movies.

https://elifesciences.org/articles/50979#fig5video1

**Figure 5—video 2.** Molecular simulation of Dll1.

https://elifesciences.org/articles/50979#fig5video2

**Figure 5—video 3.** Molecular simulation of Dll4-PP.

https://elifesciences.org/articles/50979#fig5video3

understand the correlation between molecular events and biological responses. In this study, we showed their functional and structural difference for the T cell induction in which Notch-NotchL interaction only occurs between the cells with *trans*-activation, which reflected their affinities.

The MNNL domain is known to be essential for triggering Notch signaling in the immobilized short fragment (*Andrawes et al., 2013*; *Liu et al., 2017*; *Shimizu et al., 1999*). The MNNL domain is conserved between the two NotchL families, and missense mutations were found in Alagille syndrome (*Chillakuri et al., 2012*). Moreover, Jag1 exhibited structural similarity to the C2 domain of protein kinase C, which bears phospholipid-binding properties in a calcium-dependent manner and is also necessary for efficient Notch activation (*Chillakuri et al., 2012*). Recent study reported that the MNNL of Dll4 binds directly to Notch1 (*Luca et al., 2015*), while the similarity to PKC is not detected in those of Dll ligands (*Luca et al., 2015*; *Kershaw et al., 2015*), raising the question of how MNNL contributes to the binding to Notch. Here, we showed that the MNNL domain of Dll4— that contains the loop structure with disulfide bond (C-C loop) with a wide range of motion—was critical for binding to the Notch receptor and activation of the Notch pathway, a process that results in the promotion of T cell development in vitro. The Phe[65] and His[64] residues in the C-C loop partly comprise the interface that binds to Notch1, and directly interacts with the O-Fuc moiety on Thr[466] and Leu[468]/Ile[477] residues in the EGF12 domain of Notch1 (*Luca et al., 2015*). The wide motion ensured by the flexible C-C loop structure seems to contribute to a more effective interaction with Notch1 EGF12. The actual signal transduction as assessed by luciferase reporter assay, was more affected by the substitution and insertion of unique proline residues in MNNL domain of Dll4, compared to the simple binding to Notch. This reduced signal transduction suggests that the motion of the C-C loop serves to generate the pulling force for the transendocytosis of the Notch extracellular domain. (*Weinmaster and Fischer, 2011*; *Musse et al., 2012*). The Dll4 mutant with Dll1-derived MNNL was less effective than that with two proline residues, suggesting that other part(s) of MNNL play a role in the triggering of Notch signaling.

It has been recently shown that in addition to MNNL and DSL domains, the 3rd EGF-like repeat is essential for the distinctive function of Dll ligands (*Tveriakhina et al., 2018*). We have confirmed that the 3rd EGF-like repeat of Dll4, but not of Dll1, plays an important role in signal transduction (unpublished data). In contrast to Dll1, which approaches the Notch receptor via the DSL domain and the 1st and 2nd EGF-like repeat with the DOS motif, Dll4 seems to access Notch via the MNNL and DSL domains, supported by the 3rd EGF-like repeat. The structural cooperation among these regions should be further investigated for the precise understanding of Notch-NotchL interaction.

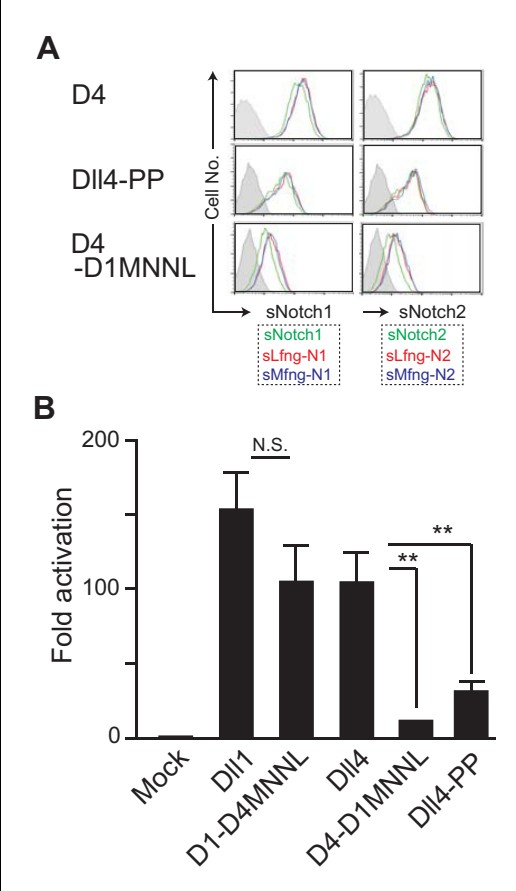

**Figure 6.** Effect of the mutation in the loop structure of MNNL domain of Dll4. (**A**) Binding activity with the soluble Notch1 and Notch2 was detected by flow cytometry as shown in *Figure 4B*. (**B**) Inducing activity of Notch signaling in vitro was examined by the co-cultures of NotchL and Notch1/Lfng transfectants as shown in *Figure 3B*. The fold activation against the control was calculated from each sample (mean ± SD, n = 3; **, p<0.01; N.S.: not significant; unpaired Student's *t*-test). Data represents three independent experiments. Expression of NotchLs were monitored by the GFP expression, the intracellular staining with anti-HA mAb and the surface staining with HRJ1-5 or anti-Dll4 mAbs (*Figure 6—figure supplement 1*).

The online version of this article includes the following source data and figure supplement(s) for figure 6:

**Source data 1.** Raw data (Fold activation) of luciferase activity used to generate the graph in *Figure 6B*.

**Figure supplement 1.** NotchLs transfectants used in *Figure 6* were monitored by the expression of GFP and the staining with anti-HA, anti-Dll4 or HRJ1-5 mAbs.

# Materials and methods

## Plasmids and constructs

We cloned cDNAs encoding the swapping mutants of Dll1 and Dll4 as below: D1-D4DSL, *Dll1* cDNA was digested by SalI/ScaI, removing the DSL region, and ligated with synthetic Dll4-derived cDNA encoding DSL domain flanked by 5' SalI and 3' ScaI sites; *D4-D1DSL*, *Dll4* cDNA was digested by AflII/ScaI, removing the DSL region, and ligated with synthetic Dll1-derived cDNA encoding DSL domain flanked by 5' AflII and 3' ScaI sites; *D1-D4E1-2*, *Dll4* cDNA was digested by ScaI/Bst1107I, removing the 1st/2nd EGF region, and ligated with synthetic Dll4-derived cDNA encoding the 1st/2nd EGF region flanked by 5' ScaI and 3' Bst1107I sites; *D4-D1E1-2*, *Dll4* cDNA was digested by ScaI/BglII, removing the 1st/2nd EGF region, and ligated with synthetic Dll1-derived the 1st/2nd EGF region flanked by 5' ScaI and 3' BglII sites; *D1-D4MNNL*, *Dll1* cDNA was digested by EcoRI (5' terminal as cloning site) and ScaI, removing MNNL and DSL regions, and ligated with EcoRI/ScaI fragment from D4-D1DSL encoding Dll4-derived MNNL and Dll1-derived DSL domains; *D4-D1MNNL*, *Dll4* cDNA was digested by EcoRI and ScaI, removing MNNL and DSL regions, and ligated with EcoRI/ScaI fragment from D1-D4DSL encoding Dll1-derived MNNL and Dll4-derived DSL domains. All cDNAs and variants for NotchLs were cloned into the MIGR1 retrovirus vector (*Pui et al., 1999*).

## Establishment of BM-derived mesenchymal cells expressing NotchLs

For establishment of BM-derived mesenchymal cell lines (VF) expressing intact NotchLs or their variants, retroviruses encoding NotchLs and mutants were obtained after transfection into the Plat-E ecotropic packaging cell line as described previously (*Abe et al., 2010*). Retrovirus-infected cells were collected 48 hr after the infection and analyzed for GFP expression by flow cytometry. GFP-positive cells were obtained by sorting, giving rise to a >99% pure population. Expression of Notch ligands or chimeric molecules on the cell surface was detected by staining with anti-Dll1, anti-Dll4 mAbs or HRJ1-5 mAb recognizing the DOS motif at the E1-2 region as described previously (*Abe et al., 2010*; *Moriyama et al., 2008*), or by intracellular staining with anti-HA mAb (Cell Signaling Technology, Danvers, MA). An isotype control of biotinylated hamster IgG and rabbit IgG were purchased from BD Biosciences (Tokyo, Japan) and Cell Signaling Technology, respectively.

For serial induction of Dll1 and Dll4, lentivirus encoding those constructed in Retro-XTM Tet-Off Advanced Inducible Expression System (Takara Bio Inc, Shiga, Japan) were prepared and infected into OP9 stromal cells. These OP9 transfectants were treated with Dox (0.03 ~ 0.1 ng/ml) to suppress the full expression of Dll molecules, and their amounts of the expression were quantitatively determined by flow cytometry with the intracellular anti-HA staining.

## Co-culture assay with stromal cells

In vitro cultures of lineage marker-negative c-kit-positive fetal liver (FL) cells with stromal cells were described previously (*Hozumi et al., 2003*; *Abe et al., 2010*). Briefly, lineage markers- (Lin; TER119, CD19, Mac-1, Gr-1) negative, c-kit-positive cells were isolated from E15.5 embryos and plated at 1 $\times$ 10$^4$ cells on a monolayer of OP9 for T cell induction into CD25$^+$ DN3 stage with serial expression of Dll for 7 days in the presence of 5 ng/ml murine IL7 (Peprotech, London, UK) and 5 ng/ml human Flt3L (Peprotech) in 24-well culture dishes or 5 $\times$ 10$^2$ cells on OP9 expressing Dll or its swapping mutants for T cell induction into DP stage for 13 days in the presence of 1 ng/ml IL7 and 5 ng/ml Flt3L. After cultures, growing cells were collected and analyzed for surface markers for 7 days (Lineage markers: Gr1, CD11b, CD11c, DX5 and ST2; CD45, Thy1.2, CD19, CD44 and CD25) or 13 days (CD45, CD19, Thy1.2, CD4 and CD8) cultures by flow cytometry. OP9 cell line was obtained from Dr. Yokoyama who published the report using this cell line (*Yokoyama et al., 2013*). We have confirmed the micoplasma-free status before the experiments.

## Antibodies and flow cytometry

FITC- and APC-CD19, CD11b, GR-1, FITC-TER119, FITC-CD44, PE-Dll1, Dll4, APC-c-kit, CD11c, B220, ST2, PE/Cy7-CD4, APC/Cy7-Thy1.2 and CD19 mAbs were all purchased from BioLegend (San Diego, CA). PE-CD19, PerCP/Cy5.5-CD25, APC-CD8, DX5, PDGFRα and PE/Cy7-CD45 mAbs were purchased from Thermo Fisher Scientific (Tokyo, Japan). Anti-NotchLs mAb, HRJ1-5, was established by immunization of Armenian hamsters with a recombinant rat Jagged1-human Fc chimeric protein (34) (R and D Systems Inc, Minneapolis, MN). Expression of NotchLs assessed by HRJ1-5 was detected by biotinylated goat anti-hamster IgG Ab (Thermo Fisher Scientific) and PE-conjugated streptavidin (BD Bioscience). Intracellular staining of HA-labeled Dll1 or Dll4 was detected by rabbit anti-HA mAb (Cell Signaling Technology) and DyLight649-conjugated donkey anti-rabbit IgG Ab (BioLegend). Expression of cell surface markers was analyzed by flow cytometry with a FACSVerse (BD Biosciences).

## Establishment and analysis of conditional dll transgenic mice

The basal targeting vector for *Rosa26* locus, ES cells with Rosa26 acceptor (IDG26.10–3) and the expression vector of PhiC31 integrase were kindly provided by Dr. R. Kühn (*Hitz et al., 2007*) (GSF National Research Center, Munich, Germany). Murine Dll1 or *Dll4* cDNA was cloned into EcoRI site of the modified targeting vector with CAG promoter and floxed GFP cassette, and introduced into ES cells bearing Rosa26 acceptor allele with the expression vector of PhiC31 integrase, followed by selection with G418. Resistant colonies were isolated and analyzed for cassette exchange by PCR with P1 and P2 or P3 primers (*Figure 1—figure supplement 1*) and by Southern blotting of BamHI-digested genomic DNA using *GFP* cDNA fragment as probe. Almost all clones showed only 8.5 kb band, indicating correct cassette exchange. Positive ES clones containing one copy of targeting construct were injected into C57BL/6 blastocysts. Resulting chimeras were bred to C57BL/6 or RosaC-reER mice (*Seibler, 2003*) and offspring were checked for germline transmission as Cre-dependent inducible Dll1 (iD1) or Dll4 (iD4) transgenic mice.

The expression of GFP in BM cells obtained from RosaCreER$^+$ iD1 and iD4 mice were analyzed with staining of CD45 and PDGFRα by flow cytometry. These mice were daily given tamoxifen (2 mg, ip; Sigma-Aldrich Japan, Tokyo, Japan) fourth times. Then, four weeks later, cells obtained from the BM were analyzed by flow cytometry. These Tg mice showed differences in the expression level of GFP. The knock-in strategy ensured the insertion of one copy of the GOI into the identical locus, and we were able to observe the same level of GFP expression in every ES cell clone obtained. Thus, we hypothesized that during or after the developmental processes of murine embryos, this locus may be regulated via some cis-element(s) included in the cDNAs, especially the 5' UTR (~300 kb in Dll4,~50 bp in Dll1) sequences.

All mouse experiments were approved by the Animal Experimentation Committee (Tokai University, Isehara, Japan).

## Binding activity of NotchLs to soluble Notch1 and Notch2

Expression vectors (pTracerCMV, Thermo Fisher Scientific) encoding murine Notch1- and Notch2-human IgG1 chimeric proteins were provided by Dr. S. Chiba (Tsukuba Univ., Japan), and the protein was collected from culture supernatant of stable transfectants of CHO cells as shown previously (*Shimizu et al., 1999*). All transfectants of NotchLs were incubated with the above supernatant for 30 min at RT, then treated with PE anti-human IgG Ab (Rockland), and analyzed by flow cytometry to detect the binding with soluble Notch proteins.

## Transient reporter assay

Reporter assays were carried out by the transient transfection of reporter plasmids TP1-luciferase (pGa981-6, including six copies of RBPJk binding sites, constructed by Dr. L. Strobl *Minoguchi et al., 1997*) and pRL-TK (Promega, Tokyo, Japan) into N1/3T3, which was established as a stable transfectant with pTracer-CMV (Invitrogen, Carlsbad, CA) encoding mouse Notch1. Each reporter plasmid (50:1) was cotransfected into $5 \times 10^4$ cells in 24-well plates by a liposome-based method (Transfast, Promega) according to the manufacturer's instructions. Following 24 hr culture after transfection, target cells were detached by trypsinization and co-cultured with stimulators ($5 \times 10^4$, VF cells expressing NotchLs) for 40 hr. Cell lysates from the mixtures of two kinds of cells were then used for the luciferase assay. Lfng-N1/3T3 was generated by the infection of retroviruses encoding Lfng (*Abe et al., 2010*).

## In silico analysis

The construction of the three-dimensional structure of Dll/Nothc1 complexes were performed with MOE, version 2016.08 (CCG Inc, Montreal, Canada) based on the Brookhaven Protein Databank 4XLW (Dll4 and Dll4-PP) and 4XBM (Dll1). Molecular mechanics calculations for these complexes were performed to prepare the initial structures for the molecular dynamics (MD) simulations using the Amber99 force field in MOE. MD simulations for these complexes were performed using the program AMBER 14 (http://ambermd.org/) with the AMBER force field and the modified TIP3P every 2.0 fs. The non-bonded interaction energy in terms of electrostatic and Van der Waal's between Dll/Notch1 complexes were calculated by NAMD Energy plug-in in VMD. The root-mean-squared deviation (RMSD) of the C-C loop in the MNNL domain of Dll (Dll1, Dll4, Dll4-PP) were analyzed by CPPTRAJ in AMBER 14.

## Acknowledgements

We thank Drs. R Kühn, S Chiba for providing the basal targeting vector for *Rosa26* locus, ES cells with Rosa26 acceptor and the expression vector of PhiC31 integrase, and soluble Notch1/Notch2 constructs; Dr. N Abe, Mr. S Ochiai for technical assistance and Drs. M Ito and N Hirayama for valuable discussion and comments for this manuscripts. We would like to thank Editage for English language editing.

## Additional information

### Funding

| Funder | Grant reference number | Author |
|---|---|---|
| Japan Society for the Promotion of Science | 16K08848 | Katsuto Hozumi |
| Ministry of Education, Culture, Sports, Science, and Technology | 22021040 | Katsuto Hozumi |

The funders had no role in study design, data collection and interpretation, or the decision to submit the work for publication.

## Author contributions
Ken-ichi Hirano, Akiko Suganami, Formal analysis, Investigation, Visualization; Yutaka Tamura, Data curation, Software, Validation, Methodology; Hideo Yagita, Resources; Sonoko Habu, Resources, Supervision; Motoo Kitagawa, Conceptualization, Validation, Methodology; Takehito Sato, Conceptualization, Validation; Katsuto Hozumi, Conceptualization, Formal analysis, Supervision, Funding acquisition, Validation, Visualization, Project administration

## Author ORCIDs
Katsuto Hozumi (iD) https://orcid.org/0000-0002-7685-6927

## Ethics
Animal experimentation: All animal experiments were performed under protocols approved by the Animal Experimentation Committee of Tokai University (Approval No.: 165015, 171002, 182026, 193040), which is further monitored by the Animal Experimentation Evaluation Committee of Tokai University with researcher for Humanities/Sociology and external expert.

## Decision letter and Author response
Decision letter https://doi.org/10.7554/eLife.50979.sa1
Author response https://doi.org/10.7554/eLife.50979.sa2

# Additional files

## Supplementary files
• Supplementary file 1. Key resources table.

• Transparent reporting form

## Data availability
All data generated or analysed during this study are included in the manuscript and supporting files. Source data files have been provided for Figures 2, 3, 4 and 6.

The following previously published datasets were used:

| Author(s) | Year | Dataset title | Dataset URL | Database and Identifier |
|---|---|---|---|---|
| Kershaw NJ, Church NL, Griffin MDW, Luo CS, Adams TE, Burgess AW | 2015 | X-ray crystal structure of Notch ligand Delta-like 1 | https://www.rcsb.org/structure/4XBM | RCSB Protein Data Bank, 4XBM |
| Luca VC, Jude KM, Pierce NW, Nachury MV, Fischer S, Garcia KC | 2015 | Complex of Notch1 (EGF11-13) bound to Delta-like 4 (N-EGF2) | https://www.rcsb.org/structure/4XLW | RCSB Protein Data Bank, 4XLW |

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
