## [Decision Letter]

**Acceptance summary:**

This paper focuses on the structural differences between Δ-like 1 (DLL1) and Δ-like 4 (DLL4) as ligands for Notch1 in T-cell development. The authors found that ectopic expression of DLL4 but not DLL1 induced T cell development in the bone marrow. Furthermore, they demonstrated that the MNNL domain of DLL4 is necessary for the Notch signaling and that the flexible movement of the C-C loop in the MNNL domain of DLL4 was important for binding and function. These findings are valuable and novel in regard to molecular selectivity. In the revised manuscript, authors provided a good overview of the previous work and explained the unique points of the current study.

**Decision letter after peer review:**

Thank you for submitting your article "Δ-like 1 and Δ-like 4 differently require their extracellular domains for triggering Notch signaling" for consideration by *eLife*. Your article has been reviewed by three peer reviewers, one of whom is a member of our Board of Reviewing Editors, and the evaluation has been overseen by Satyajit Rath as the Senior Editor.

The reviewers have discussed the reviews with one another and the Reviewing Editor has drafted this decision to help you prepare a revised submission.

Summary:

This paper from a pioneer in the field of Notch regulation of T-cell development focuses on the structural differences between DLL1 and DLL4 as ligands for Notch1. They first found that thymus immature T cells only appeared in the BM with exogenous DLL4 more efficiently DLL1. Second, more importantly, they demonstrated that the MNNL domain of DLL4 is necessary for the Notch signaling and that the flexible movement of the C-C loop in the MNNL domain of DLL4 was important for binding and function. These findings are valuable and novel in regard to molecular selectivity.

Essential revisions:

There are three points.

First, in regard to writing, authors did not fairly refer similar Tveriakhina's paper. Authors should write more explicitly with this paper's results, and more importantly, they should clearly mention which points are still not unanswered, which has been tackled by this manuscript.

Second, thymus in vivo results are not strongest part of this manuscript. The experiments in Figure 1 supports Figure 2, but the differences in expression level between the DLL4 expressing transgene and the DLL1 expressing transgene, in the hematopietic cells and stromal cells, make the results difficult to interpret. Authors do not need to remove these results, but they should discuss the results more carefully in light of this expression difference.

Finally, it is very important to quantitate the expression levels of the different chimeras in the main in vitro experiments, Figure 3 and Figure 4. The binding to Notch1 by a cell is hard to interpret unless we know how strongly each of the different chimeras is expressed on that cell.

[Editors' note: further revisions were suggested prior to acceptance, as described below.]

Thank you for resubmitting your work entitled "Δ-like 1 and Δ-like 4 differently require their extracellular domains for triggering Notch signaling in mice" for further consideration by *eLife*. Your revised article has been evaluated by Satyajit Rath (Senior Editor) and a Reviewing Editor.

The manuscript has been improved but there are some remaining issues that need to be addressed before acceptance, as outlined below:

As seen in the below reviewers' comments, reviewers request no more additional experiments, but do request authors to carefully re-write the manuscript, particularly from the following points.

1) Emphasize the unique points of this study, compared with previous reports (for instance, the results of Tveriakhina et al).

2) Describe somewhere that there could be more contribution of additional domains to DII4 vs Dll1 affinity for Notch1 that would be seen under conditions more in the middle of the dynamic range.

3) It is not always clear to the readers what alternative hypotheses can be ruled out at each point.

4) Emphasize the implications of the new supplementary data.

5) It is also not clear in the text with distinguishing between effects that are clear and effects that are not.

Reviewer 1:

I can see that the revised version has dealt with some points raised by reviewers, and the manuscript has been improved clearly. However, their explanation was not adequately clarified their advantage. Advance of this manuscript would be a role of E3 domain in comparison with the Tveriakhina's work, and the main point is that the reduced number of Pro residues in a C-C loop of the MNNL domain explain the advantage of DLL4 as ligands for Notch1. It will be more understandable as a clear advantage if the authors emphasize the specific points about the previously non-covered part of the DLL4 function.

Reviewer 2:

The authors have definitely improved the manuscript, and their addition of the new supplementary figures showing measurements of expression levels is particularly welcome. The work is quite interesting and suggests There is still a concern about the way Figure 3 was done and the way it was interpreted. However, this is not the most important figure in the paper, and on balance, I believe that the paper's remaining issues can be dealt with by changes in the writing. These issues are: (1) in the Discussion, still not fully explaining how the results in this paper specifically relate to the results reported by Tveriakhina et al; (2) concern about trying to compare affinities of binding under conditions that are apparently saturating; (3) needing more guidance for readers to understand how the newly included experiments successfully rule out some alternative explanations while favoring the explanations of the authors.

1) The Discussion has apparently not been updated since the previous submission, but it seemed important to include some kind of summary of how far the results of Tveriakhina et al. foreshadowed this paper, and how far this paper went beyond that previous report.

2) I thank the authors for explaining that they were trying to achieve such high levels of overexpression of the Notch ligands that they would saturate the Notch receptors in Figures 3 and 4. However, this is not a standard way to measure affinity differences, because the binding difference that could be due to affinity is overwhelmed by the high level of ligand. This is a problem with the way the experiments in Figures 3-4 are interpreted in terms of "affinity". In fact, the authors see no difference between wildtype Dll1 and Dll4 in the experiments of Figures 3B and 4C in the presence of Lfng, despite the fact that they have previously shown very different affinities in Figure 2. Therefore it is not clear why this data is highlighted as a way to find out what parts of the protein account for the stronger binding.

It is surprising but fortunate for the authors that the main effects seen in the chimeras are to reduce binding, even from this extremely saturated level, when the Dll1-Dll4-DSL-E1,2 chimeras are used. In fact, stronger evidence for a true binding affinity difference is seen under different conditions (soluble Notch or soluble Δ) as shown in Figure 3—figure supplement 1, panels A and B. This deserves much more emphasis and should be pointed out as a separate finding on its own. But as a caveat, please also acknowledge somewhere that there could be more contribution of additional domains to Dll4 vs. Dll1 affinity for Notch1 that would be seen under conditions more in the middle of the dynamic range.

3) In the central experiments describing functional changes of the chimeras compared to wildtype, it is not always clear to the reader what alternative hypotheses can be ruled out at each point. Unfortunately, although there is valuable new evidence to support the authors' arguments in the new supplementary figure panels, very often these panels are never mentioned in the text.

4) There are still some problems in the text with distinguishing between effects that are clear and effects that are not. Figure 3 and its supplements remain a little unsatisfying because there is no independent way to compare the levels of the Notch ligand proteins expressed among all the constructs in this experiment. (In this regard, Figure 4 and Figure 6, where the HA tag is used, are much more convincing.) However, it is still possible to help guide the reader through Figure 3. For example, one construct, D1-D4-E1-2, is poorly expressed at the construct reporter level and there is no evidence presented in the paper that this particular chimeric protein is stably expressed at all (Figure 3, Figure 3—figure supplement 1, Figure 3—figure supplement 2). In a case like this, it is possible that the lack of activity is due to lack of stable protein. However, the D1-D4DSL-E1-2 construct is at least expressed well transcriptionally (Figure 3—figure supplement 2), and this construct is equally nonfunctional. It would help the reader if the phenotype of D1-D4DSL-E1-2 were highlighted more, to emphasize the loss of function where the evidence is stronger. More important is the fact that the swap of D1-E1-2 into Dll4 actually enhances its activity instead of diminishing it a very surprising result. Here also the binding evidence and expression evidence are more convincing, but again need to be emphasized more.

---

## [Author Response]

Essential revisions:There are three points.First, in regard to writing, authors did not fairly refer similar Tveriakhina's paper. Authors should write more explicitly with this paper's results, and more importantly, they should clearly mention which points are still not unanswered, which has been tackled by this manuscript.

In accordance with your comments, we have introduced an explanation addressing the findings and unanswered points in Tveriakhina's paper in the Introduction. Moreover, we have added a further clarified our findings in the following paragraph, which are in contrast to the results reported by Tveriakhina et al.

*Second, thymus* in vivo *results are not strongest part of this manuscript. The experiments in Figure 1 supports Figure 2, but the differences in expression level between the DLL4 expressing transgene and the DLL1 expressing transgene, in the hematopietic cells and stromal cells, make the results difficult to interpret. Authors do not need to remove these results, but they should discuss the results more carefully in light of this expression difference.*

We have deliberated as to why the Dll1/4 Tg mice demonstrated the differences in expression levels. However, a definite explanation has not been reached. The knock-in strategy ensures the insertion of one copy of GOI into the identical locus, and we observed the same intensity of GFP expression in every ES cell clone obtained (please refer to Materials and methods). Thus, we postulate that during or after the developmental processes in murine embryos this locus was regulated via some cis-element(s) included in the cDNAs, especially 5’ UTR (~300 kb in Dll4, ~50bp in Dll1) sequences. This estimation has been added to the revised Material and methods section.

In this study, we aimed to confirm the superiority of Dll4 over Dll1 for the induction of T lymphopoiesis. Using these Tg mice, we successfully emphasized their differences in vivo. We have suitably revised the explanation in the Results section.

*Finally, it is very important to quantitate the expression levels of the different chimeras in the main* in vitro *experiments, Figure 3 and Figure 4. The binding to Notch1 by a cell is hard to interpret unless we know how strongly each of the different chimeras is expressed on that cell.*

We prepared every NotchL-transfectant by the repeated infection with retrovirus encoding NotchL chimeras and ensured their overexpression, monitored by the expression of GFP. In the revised manuscript, we have provided additional supplemental figures (for Figures 3, 4 and 6; introduced in each Figure legend) with their comparable fluorescence intensities. Some transfectants demonstrated lower expression of GFP; however, the protein levels of NotchLs (discuss later, monitored by intracellular HA staining) were comparable. Hence, we hypothesize that there seems to be a GFP threshold level, which ensures the saturating expression of the NotchL protein.

In case of transfectants shown in Figures 4 and 6, critical to obtain our experimental conclusion, we evaluated their expression using the intracellular staining with anti-HA mAb, assessing the total amount of the NotchL, and by the surface staining with anti-Dll4 mAb recognizing mainly the DSL region of Dll4 or with HRJ1-5 mAb recognizing the DOS motif at E1-2 region, detecting the surface expression of NotchL (Figure 4—figure supplement 1 and Figure 6—figure supplement 1). The NotchL expression in all transfectants was comparable between each Dll1/4 and Dll1/4-based chimeras; however, D4-D1MNNL demonstrated a lower expression than Dll4 as observed by anti-Dll4 mAb staining. As their protein levels evaluated by HA-stating were comparable, the affinity to anti-Dll4 mAb seemed to be affected by the replacement of the MNNL region. However, as the expression level was identical between Dll4 and D4-PP, we believe that our conclusion is accurate.

[Editors' note: further revisions were suggested prior to acceptance, as described below.]

The manuscript has been improved but there are some remaining issues that need to be addressed before acceptance, as outlined below:As seen in the below reviewers' comments, reviewers request no more additional experiments, but do request authors to carefully re-write the manuscript, particularly from the following points.1) Emphasize the unique points of this study, compared with previous reports (for instance, the results of Tveriakhina et al).

We add the new second paragraph in Discussion section explaining the unique points of this study compared with previous papers, especially Tveriakhina's paper.

2) Describe somewhere that there could be more contribution of additional domains to DII4 vs Dll1 affinity for Notch1 that would be seen under conditions more in the middle of the dynamic range.

We mention the limit of our experiment in the first paragraph in Discussion section as reviewer 2 indicated.

3) It is not always clear to the readers what alternative hypotheses can be ruled out at each point.

According to the suggestion of reviewer 2, we declare the implication of the reporter assay in Results section. In addition, we remove the word “affinity” in the explanation of our results, because it is difficult to evaluate in these experiments as indicated by reviewer 2.

4) Emphasize the implications of the new supplementary data.

We emphasize the implication of the supplemental figures of Figures 3, 4 and 6 as reviewer 2 suggested.

5) It is also not clear in the text with distinguishing between effects that are clear and effects that are not.

We add the explanation of Figure 3—figure supplement 2 in Results section as reviewer 2 suggested.